# Measurement report: Analysis of aerosol optical depth variation at Zhongshan Station in Antarctica

Lijing Chen[1,2], Lei Zhang[*1], Yong She[2], Zhaoliang Zeng[1], Yu Zheng[1], Biao Tian[1], Wenqian Zhang[1], Zhaohui Liu[3], Huizheng Che[1], Minghu Ding[1]

[1] State Key Laboratory of Severe Weather, Chinese Academy of Meteorological Sciences, Beijing, 100081, China.

[2] Chengdu University of Information Technology, Chengdu, 610103, China.

[3] Polar Surveying and Mapping Engineering Center of Heilongjiang Administration of Surveying, Mapping and Geoinformation, Harbin 150081, China

*Correspondence:* Minghu Ding (dingminghu@foxmail.com), Huizheng Che (chehz@cma.gov.cn)

Three key findings:

- The AOD level over Zhongshan Station in Antarctica is low in summer and high in winter. AE indicates the dominance of fine (coarse) aerosols in summer (winter).

- In winter and spring, high AOD values are related to the increase of coarse mode particles, while in summer and autumn, high AOD values may be related to the growth of fine mode particles.

- AOD varied inversely with wind speed and showed an insignificant positive correlation with temperature but a significant negative correlation with relative humidity.

**Abstract:** Our understanding of aerosol optical depth (AOD) in Antarctica remains limited due to the scarcity of ground observation stations and limited daylight days. Utilizing data from the CE318-T photometer spanning from January 2020 to April 2023 at Zhongshan Station, we analysed the seasonal, monthly, and diurnal variations in AOD and Ångström exponent (AE). AOD median values increased from spring (0.033) to winter (0.115), while AE peaked during summer (1.010) and autumn (1.034), declining in winter (0.381), indicating a transition in dominant aerosol particle size from fine to coarse mode between summer and winter. Monthly mean AOD variation closely paralleled the proportion of AE<1, suggesting fluctuations in coarse mode particle proportions drive AOD variation. The high AOD values during winter and spring were associated with increased contribution of coarse mode particles,

---

* Lijing Chen and Lei Zhang contribute to the work equally and should be regarded as co-first authors.

while high AOD values during summer and autumn were associated with the growth of fine mode
particles. We observed a peak in AOD (~0.06) at 14:00 local time at Zhongshan Station, possibly
associated with a slight decrease in boundary layer height (BLH). Additionally, higher (lower) wind
speeds corresponded to lower (higher) AOD values, indicating the diffusion (accumulation) effect. The
temperature and AOD showed an insignificant positive correlation ($R = 0.22$, $p = 0.40$), relative humidity
exhibited a significant negative correlation with AOD ($R = -0.59$, $p = 0.02$). Backward trajectory analysis
revealed that coarse particles from the ocean predominantly contributed to high AOD daily mean values,
while fine particles on low AOD days originated mainly from the air mass over the Antarctic Plateau.
This study enhances the understanding of the optical properties and seasonal behaviors of aerosols in the
coastal Antarctic. Specifically, AOD measurements during the polar night address the lack of validation
data for winter AOD simulations. Additionally, we revealed that lower wind speeds, higher temperatures,
and lower relative humidity contribute to increased AOD at Zhongshan Station, and air masses from the
ocean significantly impact local AOD levels. These findings help us infer AOD variation patterns in the
coastal Antarctic based on meteorological changes, providing valuable insights for climate modeling in
the context of global climate change.
**1 Introduction**
Aerosols play an important role in impacting the climate system by absorbing and scattering solar
radiation (Li et al., 2022). Antarctica, considered one of the most pristine lands, serves as an ideal
background area for evaluating the climate effects of aerosols (Kamra, 2022). Marine aerosols emitted
from the Southern Ocean are a primary source contributing to the aerosol load in Antarctica (Thakur,
2019). The retreat of sea ice in Antarctica is expected to escalate the release of sea salt and secondary
biogenic aerosols (Yan et al., 2020). Sea salt particles with strong scattering may produce negative
effective radiative forcing or indirect radiative effect by influencing cloud condensation nuclei within
the marine boundary layer over Antarctica (Thornhill et al., 2021; Udisti et al., 2012). However, the
heating effect of absorbent aerosols, such as black carbon (BC), may be amplified by the high surface
albedo in Antarctica (Kang et al., 2020). In recent years, there has been a notable increase in BC
concentrations in Antarctica, with BC deposition on snow and ice surfaces contributing to reduced

surface albedo and increased solar radiation absorption, subsequently accelerating snow and ice melt (Kannemadugu et al., 2023). Given the close connection between aerosol radiation effects and their optical properties (Che et al., 2024), it is necessary to investigate the optical parameters of Antarctica aerosols.

Aerosol optical depth (AOD), as a key parameters of aerosol optical properties, serves as an effective measure of aerosol load and can influence the solar radiation components (Alghoul et al., 2009). AOD observation records from Antarctica sites indicate that the values range from 0.006 to 0.220 in coastal regions and from 0.007 to 0.034 in inland regions (Kannemadugu et al., 2023; Tomasi et al., 2007, 2012; Yang et al., 2021). Typically, coastal aerosols consist primarily of coarse-mode sea salt particles during austral winter, while fine-mode particles (such as dimethyl sulfide and its oxidation product mesylate, DMS, and MSA) lead to elevated particle number concentrations in summer (20-100 times higher than in winter) (Lachlan-Cope et al., 2020; Shaw, 1979). Conversely, aerosols over the Antarctic Plateau predominantly comprise fine-mode particles of non-sea-salt sulfate (NSS) and DMS (Harder et al., 2000; Walters et al., 2019).

Additionally, particle size plays a significant role in aerosol extinction. The Ångström exponent (AE) serves as an important indicator of aerosol size, with value greater (less) than 1 indicating a predominance of fine (coarse) mode particles (Schuster et al., 2006). Weller and Lampert report that the mean AE at Neumayer Station was 1.5±0.6 and 1.2±0.5 during summer and winter, respectively, suggesting an increased contribution of fine-mode biological sulfate particles in summer (Weller and Lampert, 2008). Virkkula et al. observed higher scattering AE estimate values during summer (~1.9) and lower values during winter (~0.8) at Dome C on the Antarctic Plateau, indicating a prevalence of fine particles in summer (Virkkula et al., 2022).

Currently, the challenging environment and the limited number of daylight days per year restrict the availability of ground sites capable of obtaining adequate AOD and AE observations. Consequently, the optical properties of aerosols across large parts of Antarctica remain unexplored. To improve our comprehension of aerosol properties in Antarctica, we analyse the seasonal, monthly, and diurnal variations of AOD and AE using data obtained from the recently installed sun-sky-lunar CE318-T photometer at Zhongshan Station.

## 2 Site, Instrument, and Data

### 2.1 Site Introduction

Zhongshan Station (69°22′12″S, 76°21′49″E, 18 m a.s.l.) is located at the Larsemann Hills of Prydz Bay on the east Antarctic continent. The sun-sky-lunar CE318-T photometer is installed at Swan Ridge, northwest of the Nella fjord (Fig. 1) (Tian et al., 2022). This location experiences 54 polar days and 58 polar nights annually, with snow covering the surrounding surface during winter and revealing bare rock in summer. In this study, the austral spring, summer, autumn, and winter are referred to the season from September to November (SON), December to February of next year (DJF), March to May (MAM), and June to August (JJA), respectively. The average annual air temperature is -10 ℃, with a relative humidity of 58% and prevailing wind speeds of 6.9 $m\,s^{-1}$, primarily from the east or east-southeast direction (Ding et al., 2022).

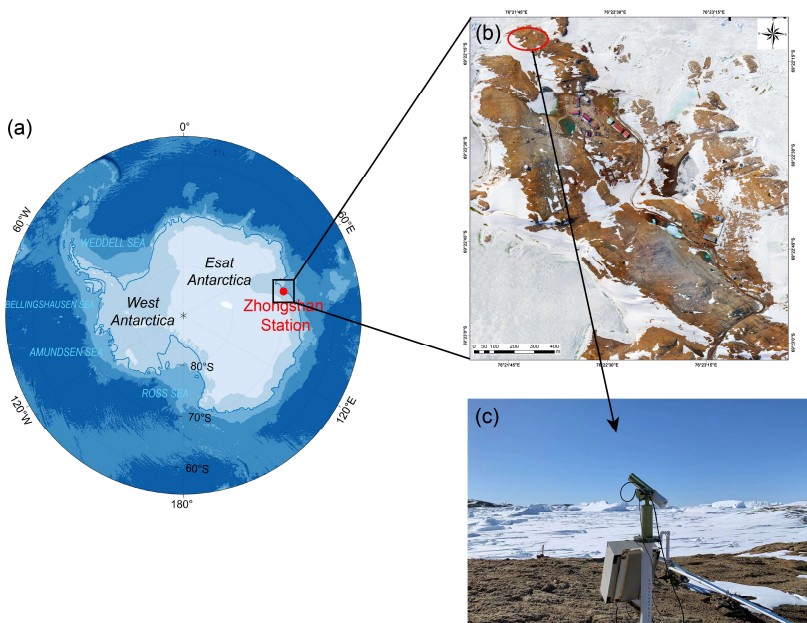

**Figure 1 (a) The location of Zhongshan Station in Antarctica, (b) the aerial view of Zhongshan Station, and (c) the sun-sky-lunar photometer CE318-T at Zhongshan Station.**

### 2.2 Instrument and Data

The AOD measurement data utilized in this study were obtained from the sun-sky-lunar CE318-T photometer, manufactured by CIMEL Electronique, France. The CE318-T is a ground-based multiband radiometer capable of inverting aerosol optical parameters by measuring the spectral data of direct solar and lunar radiation extinction and the angular distribution of sky radiances (Barreto et al., 2016).

We collected AOD level 1.5 (cloud-screened) data across various wavelengths spanning from January
2020 to April 2023 (Fig. S1). However, the operation of CE318-T in polar environment is impeded by
solar radiation and weather conditions, leading to a significant number of missing measurements.
Consequently, we categorize daily observations with less than 20 measurements and the coefficient of
dispersion (CV) exceeding 1 as invalid data, which are systematically eliminated from our analysis.
Typically, these invalid data manifest with exceedingly high AOD values, often attributed to instrument
downtime caused by factors such as precipitation or cloudy weather. Moreover, to ensure the accuracy
of AOD measurement at Zhongshan Station, we refine our data by cross-referencing station operation
records and the time series of black carbon (BC) concentrations. This process allows us to exclude AOD
data associated with significant station activities and periods of elevated BC concentrations, thereby
enhancing the reliability of our analysis. It should be noted that there are uncertainties in the AOD
measurements of CE318-T during field observations due to atmospheric conditions, instrument noise,
and calibration. It is estimated that during daytime measurements, the AOD uncertainty ranges from
0.010 to 0.021. For night-time measurements, the AOD uncertainty depends on the calibration technique
used. Specifically, when calibrated using the Moon Ratio technique, the uncertainty ranges from 0.011
to 0.019. With the application of the new Sun Ratio technique, the uncertainty for the 440 nm channel is
between 0.012 and 0.015 (0.017), while for longer wavelengths, it ranges from 0.015 to 0.021. By
employing the new Sun-Moon gain factor technique and using the Langley-calibrated instrument for
calculation of the amplification between daytime and night-time measurements, the uncertainty range is
from 0.016 to 0.019 (Barreto et al., 2016).
The meteorology data, including temperature, relative humidity, wind direction, and wind speed, were
obtained from the Zhongshan Station meteorology observatory, with the temporal resolution of 1 hour.
BLH data was obtained from ERA5 reanalysis provided by European Centre for Medium Range Weather
Forecasts (ECMWF) with the temporal and spatial resolution of 1 hour and 0.25 (latitude) $\times$ 0.25
(longitude).
The Hybrid Single-Particle Lagrangian Integrated Trajectory (HYSPLIT) model, is a comprehensive
model developed by the National Oceanic and Atmospheric Administration (NOAA) and the Air
Resources Laboratory (ARL) to calculate and analyse the source, transport, and diffusion trajectories of

atmospheric pollutants. The meteorological data used in the HYSPLIT model comes from the National

Center for Environmental Prediction (NCEP) Global Data Assimilation System (GDAS). In this study,

the HYSPLIT model is utilized to calculate the 168h backward air mass trajectory from 3 altitudes of 50,

500, and 1000 m (amsl) to Zhongshan Station.

**3 Results**

**3.1 Variation Characteristics of AOD**

From January 2020 to March 2023, the monthly mean AOD values at various wavelengths varied from

0.00 to 0.20, with the lowest values in December 2020 and the highest values in August 2022 (Fig. 2a).

The monthly mean AOD values at 500 nm ($AOD_{500\ nm}$) generally remained below 0.10, consistent with

findings by Gadhavi and Achuthan at the Maitri Station, where AOD variation fell within the range of

0.01 to 0.10 (Gadhavi and Achuthan, 2004). The annual mean $\pm$ SD (standard deviation) values of the

$AOD_{500\ nm}$ were 0.074$\pm$0.090, 0.051$\pm$0.066, 0.071$\pm$0.117, and 0.053$\pm$0.031 in 2020, 2021, 2022, and

2023, respectively (Table 1). Similarly, the annual mean $\pm$ SD values of the $AE_{440\text{-}870\ nm}$ were

1.134$\pm$0.411, 0.953$\pm$0.338, 0.883$\pm$0.374, 0.753$\pm$0.206 for the same years, respectively, suggesting that

the aerosols over Zhongshan Station were mainly dominated by fine mode particles in 2020, and coarse

mode particles in 2021, 2022, and 2023. The relationship between multi-year $AOD_{500\ nm}$ and $AE_{440\text{-}870\ nm}$

illustrates that fine mode particles are primarily concentrated in the range of $AOD_{500\ nm} < 0.1$, while high

$AOD_{500\ nm}$ values, which occur occasionally, are caused by coarse mode particles (Fig. 2b). Although

fine mode particles have a longer suspension time in the atmosphere and can efficiently scatter and absorb

sunlight, leading to lower AOD ranges, it is worth mentioning that in the coastal regions of Antarctica,

the dominant role in AOD is sometimes played by coarse mode particles. These particles, with larger

radii and higher volume concentrations, originate mainly from abundant sea salt sources. Their presence

results in increased scattering and absorption of sunlight, emphasizing the significance of coarse mode

particles in determining AOD levels in the Antarctic coastal areas (Su et al., 2022)

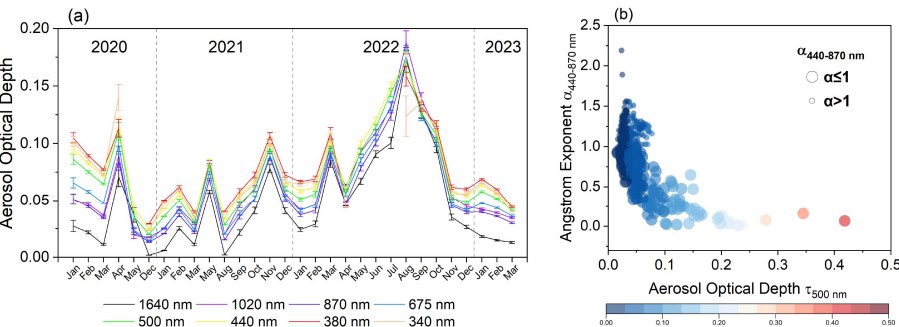


**Figure 2 (a) Monthly variation of mean aerosol optical depth at different wavelengths measured over**
**Zhongshan Station in Antarctica from 2020 to 2023. (b) Relationship between $AOD_{500\,nm}$ and $AE_{440\text{-}870\,nm}$ over**
**Zhongshan Station from 2020 to 2023.**
**Table 1 Annual mean and standard deviation of aerosol optical depth at different wavelengths and Angstrom**
**Exponent at 440-870 nm at Zhongshan Station from 2020 to 2023.**

|  | 2020 | 2021 | 2022 | 2023 |
|---|---|---|---|---|
| $AOD_{1640\,nm}$ | $0.028\pm0.102$ | $0.026\pm0.079$ | $0.050\pm0.141$ | $0.016\pm0.036$ |
| $AOD_{1020\,nm}$ | $0.049\pm0.095$ | $0.045\pm0.073$ | $0.067\pm0.131$ | $0.040\pm0.034$ |
| $AOD_{870\,nm}$ | $0.047\pm0.093$ | $0.039\pm0.070$ | $0.060\pm0.126$ | $0.037\pm0.032$ |
| $AOD_{675\,nm}$ | $0.059\pm0.091$ | $0.042\pm0.068$ | $0.063\pm0.122$ | $0.044\pm0.031$ |
| $AOD_{500\,nm}$ | $0.074\pm0.090$ | $0.051\pm0.066$ | $0.071\pm0.117$ | $0.053\pm0.031$ |
| $AOD_{440\,nm}$ | $0.081\pm0.089$ | $0.057\pm0.065$ | $0.077\pm0.116$ | $0.057\pm0.031$ |
| $AOD_{380\,nm}$ | $0.089\pm0.091$ | $0.063\pm0.065$ | $0.077\pm0.117$ | $0.061\pm0.032$ |
| $AOD_{340\,nm}$ | $0.088\pm0.095$ | $0.059\pm0.064$ | $0.073\pm0.118$ | $0.058\pm0.032$ |
| $AE_{440\text{-}870\,nm}$ | $1.134\pm0.411$ | $0.953\pm0.338$ | $0.883\pm0.374$ | $0.753\pm0.206$ |


**3.2 Seasonal and Monthly Variations in AOD and Ångström Exponent**
The seasonal variation of $AOD_{500\,nm}$ and $AE_{440\text{-}870\,nm}$ over Zhongshan Station suggests the median $AOD_{500}$
$_{nm}$ values are lower in spring (0.033), summer (0.036), and autumn (0.045), but higher in winter (0.115),
while the $AE_{440\text{-}870\,nm}$ values are 0.908, 1.010, 1.036, and 0.381, respectively (Fig. 4a). The frequency
histograms show that the highest frequency range of $AOD_{500\,nm}$ is 0.02 to 0.04 in spring, summer, and
autumn, while 0.08 to 0.12 in winter (Fig. S2). The normal fitting curves of the frequency histograms of
$AE_{440\text{-}870\text{ nm}}$ indicate that the peak in winter is in the low-value range (0.3~0.4), while the peaks in spring,
summer, and autumn are in the high-value range (1.0~1.2). To investigate the seasonal differences in
$AOD_{500\text{ nm}}$ and $AE_{440\text{-}870\text{ nm}}$, it is essential to understand the sources of air masses influencing aerosols at
Zhongshan Station. Therefore, we calculate the 168h backward trajectory once every 24h from January
2020 to December 2022, with the starting height of 500 m and the starting time of 18:00 (13:00 local
time at Zhongshan Station), and clustered by season (Fig. 3). We observed that the proportion of air
masses originating from the surrounding waters (red clusters) and the ice edge margin of ice sheet (blue
clusters) ranges from 50% to 80%, dominating throughout the year. These air masses are likely the
primary sources of local or natural aerosols in Antarctica. In contrast, the proportion of distant origin
(yellow and green clusters) is approximately 20%, but it significantly increases in autumn, reaching
around 45%. These air masses are associated with long-range transported aerosols. Therefore, the
seasonal differences in $AOD_{500\text{ nm}}$ and $AE_{440\text{-}870\text{ nm}}$ at Zhongshan Station are largely attributed to
variations in the types and concentrations of local aerosols.
The seasonal variations in AOD and AE are consistent with previous findings on sea salt aerosol
concentrations, although the mechanism behind this seasonal variation is multifaceted. Wang and Huang
et al. have indicated that higher winter wind speeds at Zhongshan Station can elevate marine source
aerosol concentrations, primarily composed of sea salt, potentially explaining the winter peak in sea salt
concentration (Hong et al., 2009; Huang et al., 2005). However, Hall and Wolff propose that the high
sea salt load correlates more with moderate wind speeds and shifts in wind direction, rather than high
wind speeds, with concentrated brine on freshly formed ice surfaces acting as a source of winter sea salt
(Hall and Wolff, 1998). Moreover, blowing snow over sea ice generates aerosols primarily made of sea
salt, contributing to the winter peak in sea salt aerosols (Frey et al., 2020). In summer, lower sea salt
concentrations lead to lower background levels of AOD, but the effect of enhanced marine biogenic
emissions on AOD may increase. In the marine boundary layer over the eastern Southern Ocean sector,
$nssSO_4^{2-}$ and MSA contribute approximately 40% of the total mass of fine aerosols (particle size < 0.56
$\mu m$) (Xu et al., 2021). Xu et al. reported the annual mean concentrations of $nssSO_4^{2-}$ and MSA at
Zhongshan Station were 0-79 $ng\ m^{-3}$ and 19-41 $ng\ m^{-3}$, respectively, with the maximum
concentrations were observed in summer (Xu et al., 2019). This increase in summer concentrations is

attributed to enhanced solar radiation, phytoplankton blooms in the polynyas releasing DMS (Zhang et al., 2015), and the DMS in the atmosphere is oxidized by radicals such as $O_3$ (significant at high latitudes), OH, and BrO in the gas phase (Boucher et al., 2003), resulting in elevated concentrations of MSA and $nssSO_4^{2-}$. The positive correlation between mean surface chlorophyll and AOD in the Southern Ocean confirmed the contribution of DMS flux to aerosol load during summer (Gabric et al., 2005).

The monthly variations in $AOD_{500\ nm}$ and $AE_{440-870\ nm}$ at Zhongshan Station suggest an opposite trend, with the mean values of $AOD_{500\ nm}$ peaking in July and $AE_{440-870\ nm}$ reaching its lowest in June (Fig. 4b). Median $AOD_{500\ nm}$ values increase slightly from January to February, followed by a decrease in March and increase continuously from March to August, reach the maximum value, then gradually decrease, and reach the minimum in November and December. The percentages of $AE_{440-870\ nm} > 1.0$ and $AE_{440-8870\ nm} < 1.0$ represent the proportion of the monthly occurrence frequency of fine and coarse mode particles (Fig. 4c). The monthly mean and median $AOD_{500\ nm}$ values are consistent with the proportion of coarse mode particles ($AE_{440-870\ nm} > 1.0$), suggesting that the variation characteristics of $AOD_{500\ nm}$ at Zhongshan Station are primarily influenced by coarse mode particles. Given that Zhongshan Station is located in the coastal area of Antarctica, it is suspected that these coarse particles may be sea salt aerosols.

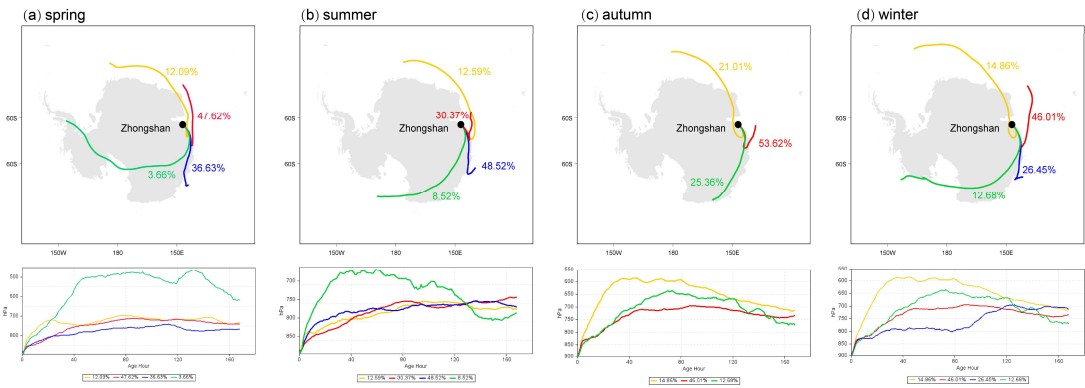

**Figure 3 Clusters of air mass backward trajectories in (a) spring, (b) summer, (c) autumn and (d) winter at Zhongshan Station from 2020 to2022.**

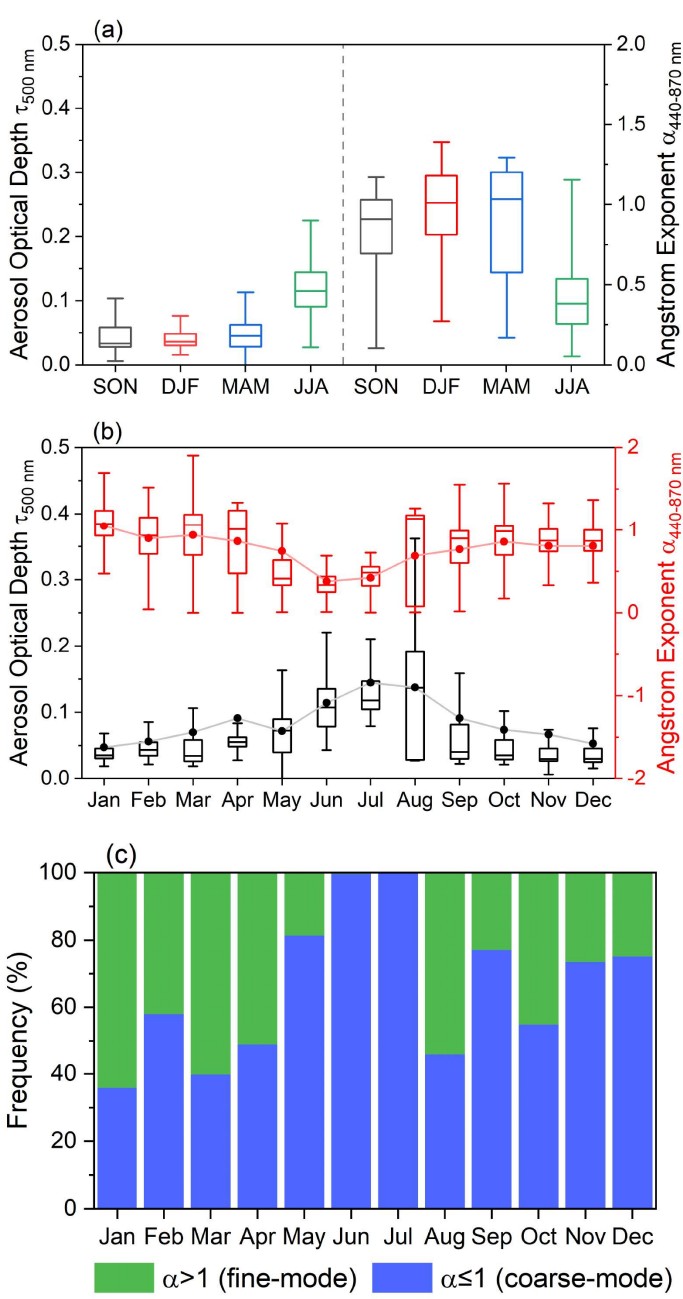

**Figure 4 (a) Seasonal variation of aerosol optical depth at 500 nm and Angstrom exponent at 440-870 nm over Zhongshan Station. For each monthly box, the central line indicates the median; and the bottom and top edges of the box indicate the 25th and 75th percentiles, respectively. (b) Variations in monthly AOD$_{500 nm}$ and AE$_{440-870 nm}$ at Zhongshan Station. For each monthly box, the central line indicates the median; the dot represents the mean; and the bottom and top edges of the box indicate the 25th and 75th percentiles, respectively. (c) Monthly percentages of Ångström exponent >1.0 (green) and Ångström exponent ≤ 1.0 (blue) at Zhongshan Station from 2020 to 2023.**

Additionally, we used a graphical method proposed by Gobbi et al (Gobbi et al., 2007), which is based on Mie calculation and correlates Ångström exponent ($\alpha$) and Ångström exponent spectral difference ($\delta\alpha$) with fine mode aerosol effective radius ($R_{eff}$) and fine mode fraction to investigate the aerosol

modification processes at Zhongshan Station in different seasons. Fig. 5 presents a schematic diagram of
the classification of aerosol types using the $\alpha$ and $\delta\alpha$ functions of a dual-mode, lognormal distribution
with refractive index $= 1.4 - 0.001i$ as reference. It is known from Jurányi and Weller' research that the
refractive index of Antarctic coastal aerosol is about 1.4, so it seems reasonable to use this reference
(Jurányi and Weller, 2019). We utilized $AOD_{440nm}$, $AOD_{675nm}$, and $AOD_{870nm}$ to calculate $\alpha_{440-675nm}$,
$\alpha_{440-870nm}$, and $a_{675-870nm}$, and then get the $\delta\alpha = \alpha_{440-675nm} - a_{675-870nm}$. The negative values of
$\delta\alpha$ indicate the dominance of fine mode aerosol, while positive values indicate the effect of two separate
particle modes (Kaufman, 1993). The solid black line represents the size of fine mode particles ($R_{eff}$),
and the dashed blue line represents the proportion of the contribution of fine mode particles to AOD ($\eta$).
In Fig. 5, increasing $AOD_{675\ nm}$ is associated with the declining $\eta$ (spring and winter) and increasing
$R_{eff}$ (summer and autumn). This indicates that higher aerosol loads in spring and winter are attributed
to increased coarse-mode particle fractions, whereas in summer and autumn are primarily associated with
the increase of fine-mode particle size. Previous studies have indicated that sea salt dominates winter
aerosols in the coastal areas of Antarctica (Hall and Wolff, 1998; Weller et al., 2008), and Xu et al
observed that the highest mean concentration of sea salt in September at Zhongshan Station, these can
explain the $\delta\alpha$ values are mainly positive in spring and winter, and $\eta$ is concentrated within the range
of less than 50% (Xu et al., 2019). In summer and autumn, apart from common sea salt aerosols ($\delta\alpha > 0$,
$\eta < 50$), the high AOD is mainly related to the particle growth such as hygroscopic growth or
condensation of fine mode aerosols ($R_{eff}: 0.10\mu m \sim 0.20\mu m$). This may be linked to the atmospheric
oxidation of (DMS) emitted by biological sources in coastal regions, or the aging process of aerosols
originating from other sources, as the rate of new particle formation and particulate matter growth in
summer is much greater than in winter in the Antarctica (Davison et al., 1996; Lachlan-Cope et al., 2020;
Weller et al., 2015).

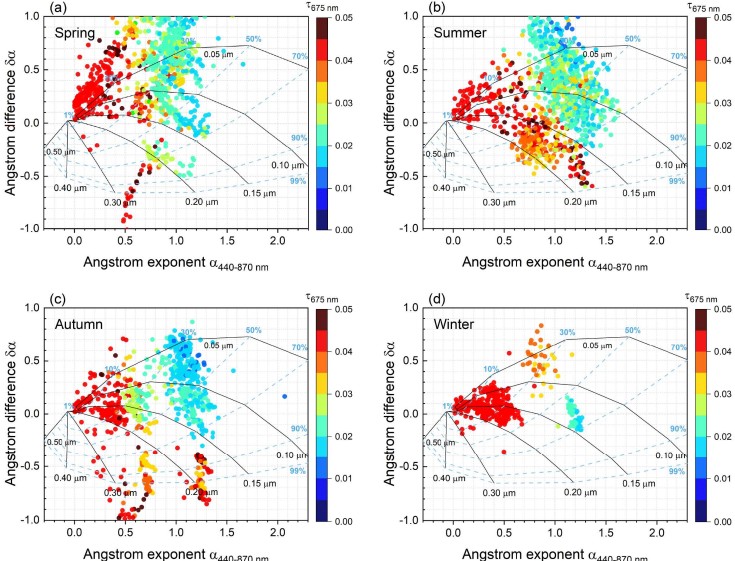


**Figure 5 Ångström exponent difference ($\delta\alpha = \alpha_{440-675\,nm} - \alpha_{675-870\,nm}$) as a function of the $\alpha_{440-870\,nm}$**
**and AOD$_{675\,nm}$ (colour scale) during (a) spring, (b) summer, (c) autumn, and (d) winter at Zhongshan Station.**
**The black lines indicate the $R_{eff}$ of fine-mode aerosols, while the blue lines correspond to fine-mode fraction**
**($\eta$).**
**3.3 Relationship between AOD, Ångström Exponent and Meteorological Conditions**
In this section, we analyse the diurnal variation characteristics of AOD$_{500\,nm}$ and AE$_{440-870\,nm}$ during
summer and explore their correlation with meteorological variables within the planetary boundary layer
(PBL), such as wind directions and speeds, temperature, and relative humidity. We calculated the diurnal
variations of AOD$_{500\,nm}$ and AE$_{440-870\,nm}$ based on observations collected at Zhongshan Station during
summer (December-February, 2020-2023), with each hourly mean containing at least one thousand
individual observations (Fig. 6). The mean AOD$_{500\,nm}$ exhibited an increase from 5:00 to 14:00 (local
time of Zhongshan Station), reaching a maximum value (0.06±0.04), and then decreased. The mean
AE$_{440-870\,nm}$ decreased from 5:00 to 12:00, reaching the lowest value (0.85±0.25), and then increased.
These results indicate that the highest aerosol load occurs at 14:00, accompanied by a larger aerosol
particle size during this period. The diurnal variation of boundary layer height (BLH) is almost consistent
with the variation of AOD$_{500\,nm}$, which is inconsistent with the general conclusion that the negative
correlation between BLH and particulate matter concentration in the mid-latitudes (Lou et al., 2019;
Miao and Liu, 2019). However, a minor decline in BLH is noticeable when the AOD$_{500\,nm}$ value reaches
its peak at 14:00. Consequently, we suspect that the weak absorption and low content of Antarctic
aerosols typically do not suffice to form an "aerosol-boundary layer" positive feedback mechanism, but
may contribute to reducing the BLH when AOD is high (Lou et al., 2019; Petäjä et al., 2016).

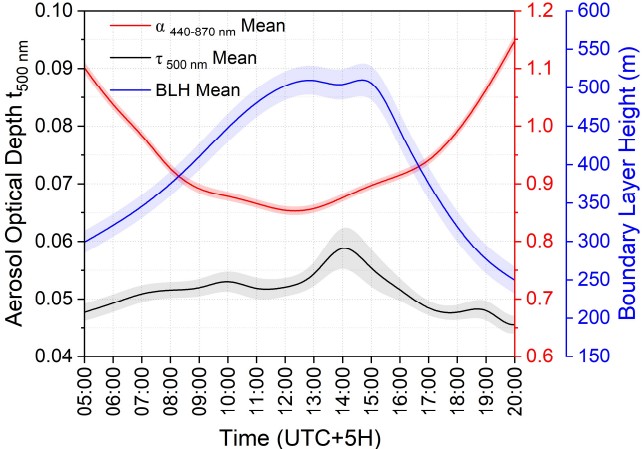


**Figure 6 Diurnal variation of AOD$_{500\ nm}$ and AE$_{440\text{-}870\ nm}$ at Zhongshan Station. The black line indicates the**
**mean of AOD$_{500\ nm}$; the red line represents the mean of AE$_{440\text{-}870\ nm}$; the blue line represents the mean of BLH.**
**The shadow represents the standard deviation of the mean.**

Moreover, the 2-minute average wind direction at Zhongshan Station mainly comes from the southeast,
with the diurnal variation of the 2-minute average wind speeds ranging from 2 to 9 m s$^{-1}$. There is a
noticeable decline in wind speeds between 5:00 and 14:00, followed by a gradual increase thereafter (Fig.
7). Given that the CE318-T is positioned westward of the main Zhongshan Station building, the eastward
winds may carry emissions originating from western stations such as Zhongshan and Progress Station.
The relationship between the diurnal variation of AOD$_{500\ nm}$ and wind speed is more obvious: AOD$_{500\ nm}$
exhibits a decline (increase) concurrent with increasing (decreasing) wind speeds. This correlation stems
from the fact that higher wind speeds facilitate the dispersion of pollutants, leading to a reduction in
AOD, and vice versa (Coccia, 2021; Liu et al., 2020; Wang et al., 2022).

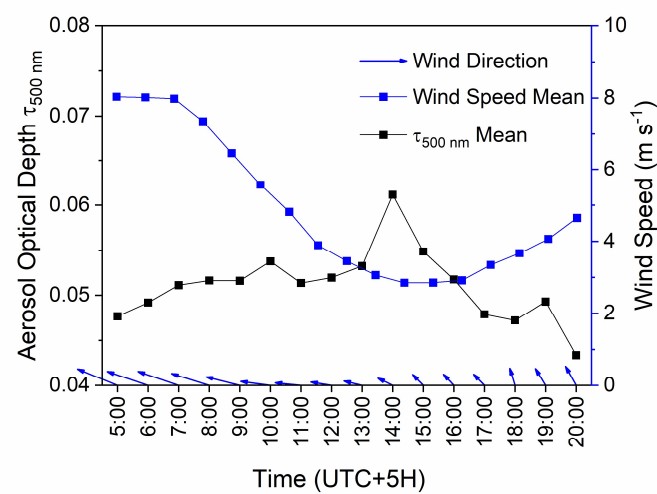

**Figure 7 Diurnal variations of 2-minute average wind direction, 2-minute average speed, and AOD$_{500\ nm}$ in summer at Zhongshan Station.**

The influence of temperature and relative humidity on aerosol parameters is relatively complex. Temperature affects aerosol particle concentration by influencing the air convection and influencing the formation and optical properties of secondary by controlling chemical transformation (Han et al., 2007; Li et al., 2020). Relative humidity affects the chemical composition, size distribution, and optical properties of aerosol particles by affecting their aqueous-phase reactions and gas-particle partitioning (Altieri et al., 2008; Ding et al., 2021; Hennigan et al., 2008; Sun et al., 2013). The diurnal variations of AOD$_{500\ nm}$, temperature, and relative humidity in summer at Zhongshan Station show that AOD$_{500\ nm}$ is positively correlated with temperature with a correlation coefficient of 0.22, and AOD$_{500\ nm}$ is negatively correlated with relative humidity with a correlation coefficient of -0.59 (Fig. 8). This indicates that rising (declining) temperature and declining (rising) relative humidity during the day may contribute to an increase (declining) in aerosol load. Previous studies have shown a positive correlation between temperature and AOD (Basharat et al., 2023). During the summer at Zhongshan Station, high temperatures may destroy the physical properties of bare rocks and promote the formation and diffusion of particulate matter, thereby increasing the aerosol load (Zhang, 2024). However, there is a study showing that higher temperatures may reduce methane sulfinic acid (MSIA) yield (Cecilia Arsene et al., 1999). Therefore, the effect of temperature on the AOD at Zhongshan Station is complex, resulting in an insignificant positive correlation. The relationship between relative humidity and AOD is inconclusive (Gautam et al., 2022), as high relative humidity may contribute to the increase of aerosol hygroscopic properties leading to an increase in AOD (Meng et al., 2024), or it may contribute to a decrease in AOD

by reducing dust particles in the air (Zhang, 2024). Therefore, the influence of temperature and relative
humidity on AOD may be related to the physicochemical properties of local aerosols and their sourcing
and sink processes.

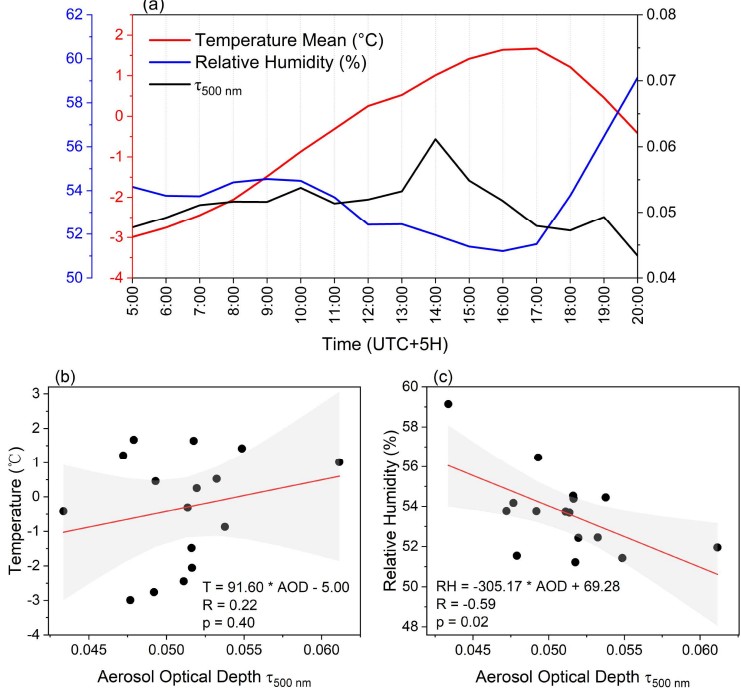


**Figure 8 (a) Diurnal variations of AOD$_{500 nm}$ (black), temperature (red), and relative humidity (blue) in**
**summer at Zhongshan Station; (b) relationship between AOD$_{500 nm}$ and temperature; and (c) relationship**
**between AOD$_{500 nm}$ and relative humidity. The red line indicates the regression line obtained by fitting a linear**
**regression, and the grey bands indicate the confidence intervals for the linear regression.**
**4 Discussion**
**4.1 Potential effects of aerosol sources on AOD levels**
Besides meteorological conditions, aerosol sources may also influence the diurnal variation
characteristics of AOD. We classified days with mean AOD below the 5th percentile as low AOD day
and those above the 95th percentile as high AOD day (Fig. S3 and Table S1). Using the HYSPLIT
backward trajectory model, we found that air masses on high AOD days primarily originated from the
ocean, whereas those on low AOD days mostly came from the interior of Antarctica (Fig. S4). The
altitudes of the backward trajectories show that during low AOD days, the air mass originating from the
ocean usually starts at a lower altitude (<1000 m), rises to a higher altitude (~2000 m) and then descends
to Zhongshan Station (2020-05-15 and 2020-12-25), while the air mass originating from the interior of
Antarctica usually starts at a higher altitude (~3000 m) and then descends to Zhongshan Station. This
indicates that particles from the Antarctic plateau or the free troposphere above the Antarctic interior are
transported to Zhongshan Station by katabatic winds. Researches show that the katabatic winds driven
by latent cooling occurring in the high-wind East Antarctic can rush the dense air from the interior plateau
to the coast (Simmons et al., 2021; Yu et al., 2020). Combined with the AE values, we can find that the
AE values of low AOD days are usually greater than 1, indicating the small particle size, thus, we suspect
that these fine particles may be $nssSO_4^{2-}$ from the Antarctic interior (Pei et al., 2021). In contrast, in
high AOD days, the air mass all originates in the ocean and usually starts at a lower altitude. The AE
values corresponding to high AOD moment on high AOD days are extremely low (<0.5), indicating that
the particle size is large, thus, we suspect that these aerosols may consist of coarse sea salt particles.
**4.2 Potential effects of aerosol particles on cloud and radiative forcing**
The optical properties of aerosols play a crucial role in their impact on radiative forcing, cloud formation,
and local climate. In our analysis of the variations in AOD and AE, we provided insights into the aerosol
loading, particle sizes, and possible formation and growth mechanisms in the atmosphere over
Zhongshan Station. During winter and spring, coarse mode particles are predominantly derived from sea
salt. Studies have shown that aerosols larger than 0.13 μm in the marine boundary layer contain sea salt,
contributing to most of the aerosol scattering and inducing cooling effects (Murphy et al., 1998).
Additionally, the size and inhomogeneity of sea salt particles are often associated with relative humidity.
Compared to remote oceans, the low relative humidity in coastal Antarctica may introduce more
inhomogeneous sea salt particles, resulting in up to a 12% change in direct radiative forcing due to
inhomogeneity (Wang et al., 2019).
However, we are particularly interested in the behaviour of aerosol particles during summer since solar
radiation is limited in winter. In summer and autumn, the increase in fine mode particles in closely related
to the release of biogenic aerosols, such as DMS, emitted by phytoplankton in the marginal ice zone.
When particles grow to a size suitable for cloud condensation nuclei or ice nucleating particles, they can
affect the formation of low-level mixed-phase clouds in coastal areas, contributing to the formation of
low-level ice clouds. At the same time, the increased number density of cloud droplets enhances cloud
reflectivity, resulting in negative radiative forcing (Satheesh and Krishna Moorthy, 2005). A recent study
revealed that in the shallow mixed-phase clouds over Antarctica, the concentrations of cloud-relevant
aerosol particles match the concentrations of ice crystals and cloud droplets (Radenz et al., 2024). the
number of particles plays a crucial role in cloud growth. Increasing particle concentration results in a
higher abundance of liquid droplets and ice crystals within clouds, which can impact cloud lifespan and
potentially influence local weather and climate. Therefore, continuous monitoring of aerosol optical
properties in coastal Antarctica is vital to improve our comprehension of aerosol radiative forcing
variations caused by changes in aerosol loading and particle size.
**5 Summary**
This study analysed the AOD and AE variations retrieved from CE318-T sun photometer data spanning
from January 2020 to April 2023 at Zhongshan Station in Antarctica. The main conclusions we draw are
as follows:
At Zhongshan Station, AOD varied from 0.00 to 0.20. Fine mode particles were predominantly found in
the lower AOD range, while higher AOD values were mainly attributed to coarse mode particles.
Seasonally, AOD exhibited a pattern of lower values in summer and higher values in winter, and the AE
displayed an opposite trend. The increases in AOD during summer and autumn may be linked to particle
growth, whereas the increases during spring and winter are associated with a decline in the fraction of
fine mode particles.
Low aerosol load over Zhongshan Station was not enough to form an "aerosol-boundary layer" positive
feedback mechanism, but the slight decrease in BLH may be related to AOD diurnal peak at 14:00.
Moreover, high (low) wind speeds facilitated pollutant dispersion (accumulation), leading to reduced
(increased) AOD. A weak positive correlation was noted between temperature and AOD (R = 0.22, p =
0.40), and a negative correlation between relative humidity and AOD (R = -0.59, p = 0.02). The
mechanisms underlying temperature and humidity's influence on aerosols remain unclear, possibly
linked to local aerosol properties at Zhongshan Station. In addition, we discuss the influence of aerosol
sources on AOD. The backward trajectories show that the air masses on high AOD days come from the
ocean, and the low AE values indicate that the particle size is larger, we speculate that the main
composition of the aerosols is sea salt. The air masses on the low AOD days mainly come from the

interior of Antarctica, and the high AE values indicate that the particle size is small. We speculate that

the katabatic winds rush the air from the Antarctic plateau to Zhongshan Station.

**Data availability**

The data included in this study can be accessed via https://zenodo.org/records/10983098. Boundary layer

height data downloaded from ECMWF ERA5 (https://www.ecmwf.int/en/forecasts/dataset/ecmwf-

reanalysis-v5). Backward trajectory of air mass and the meteorological data are obtained from NOAA

Air Resources Laboratory (https://www.ready.noaa.gov/HYSPLIT_traj.php).

**Author contributions**

The paper is a result of the lead author's research work under the supervision of MD, LZ, YS. ZZ and

YZ provided constructive comments. LZ participated in the offline discussion and made equal

contributions in responding to the review comments and revising the manuscript. MD, QW and BT

provided experimental data. ZL provided aerial photos of Zhongshan Station. LC wrote the first draft of

the paper with the help and support of all the authors. HC provided guidance for the manuscript revisions.

**Competing interests**

The contact author has declared that none of the authors has any competing interests.

**Acknowledgments**

Funding for this study was provided by the National Natural Science Foundation of China (42122047),

the National Key Research and Development Program of China (2021YFC2802504), and the Basic

Research Fund of the Chinese Academy of Meteorological Science (2023Z015&2023Z025).

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
