# Peer review of "Measurement report: Analysis of aerosol optical depth variation at Zhongshan Station in Antarctica"

_EGUsphere, 2024_

## Author Comment (AC1)

**Anonymous Referee #1:**

*This study is too simple and may be well-suited as a measurement report. Undoubtedly, the aerosols in that location are very important to study, but the paper lacks enough depth to it.*

*However, I have some minor comments*

Respond:

Thanks for your comments, we have changed the type of this manuscript to "measurement report", we hope to use the current measurement data to increase the understanding of the characteristics of aerosol optical depth at Zhongshan Station. The following are our responds to your comments, with black text for your comments and blue text for our responds. We hope our responds will address your concerns.

*"The increase in AOD during spring and winter correlates with a reduction in the fine mode fraction, whereas the increase observed in summer and autumn may be attributed to the growth and aging of fine particles." How can both increase? Please correct*

Respond:

Thanks for your questions. In Section 3.2, we use the aerosol classification method proposed by Gobbi to obtain the contribution of fine mode particles to AOD and the size of fine mode particles, and can also separate whether the increase of AOD is caused by the hygroscopicity growth of fine mode particles or by the increase of coarse mode particles. In Fig. 4, we discussed the different seasons. For example, in spring (Fig.4a), when AOD<0.03, aerosol is mainly composed of fine mode particles ($\alpha_{440-870\,nm} > 1.0$), the contribution of fine mode particles to AOD is less than 70% ($\eta < 70\%$); when AOD>0.03, aerosol is dominated by coarse mode particles ($\alpha_{440-870\,nm} < 1.0$), and the contribution of fine mode particles to AOD is less than 30% ($\eta < 30\%$). Moreover, since the particles during spring are mainly concentrated in the $\delta\alpha > 0$, this indicates that the increase of coarse mode particles is main reason for the increase of AOD in spring. In summer (Fig.4b), when AOD>0.03, aerosols mainly concentrated in the region of fine mode particle growth ($\delta\alpha < 0$), and the contribution of fine mode particles to AOD is 50% to 99% ($50\% < \eta < 99\%$), indicating a significant influence of fine mode particle

growth on AOD increase. Therefore, we believe that the main reason for the higher AOD values is different during different seasons.

We apologize for this unclear statement, which may lead to confusion for readers. The following is our change:

In winter and spring, high AOD values are related to increase of coarse mode particles, while in summer and autumn, high AOD values may be related to the growth of fine mode particles.

*"Increases in AOD during spring and winter correlated with decreases in fine mode fraction, while increases during summer and winter related to fine mode particle growth and aging." This line is very confusing with the usage of 'increase''*

Respond:

Thanks for your comment, it is helpful to improve the quality of our manuscript. The following is our change:

 The high AOD values during winter and spring were associated with increased contribution of coarse mode particles, while high AOD values during summer and autumn are associated with the growth of fine mode particles.

*The last line in abstract only talks about the origins of particles in the summer. Why specifically summer? Why not other seasons?*

Respond:

Thank you for pointing out the problem. In the discussion part, we used the air mass backward trajectory to discuss the sources of aerosol on high AOD days and low AOD days respectively. Although most of the high AOD days occur in summer, it is inaccurate to mention only summer in the abstract. The following is our change:

Backward trajectory analysis revealed that coarse particles from the ocean predominantly contributed to high AOD daily mean values, while fine particles on low AOD days originated mainly from the air mass over the Antarctic Plateau.

*The abstract is incomplete. I suggest you to add a conclusion line to your abstract as to why this study is important or how it can help others?*

Respond:

Thanks for your suggestion, it is very important for our manuscript! The following is what we added in abstract:

This study analyzed the variation characteristics of AOD on different time scales at Zhongshan Starion in Antarctica, and the observation during the polar night is helpful for us to understand the variation of AOD in winter. In addition, we provide important insights into regional AOD levels affected by meteorological conditions and aerosol sources in Antarctica.

*How is DMS found in the plateau? Does it come from transportation from ocean? But you have mentioned about katabatic winds that drive from interior to coastal*

Respond:

Thank you for your comments. In fact, we did not observe DMS particles in the Antarctic plateau. In Section 3.2, the increased concentration and growth of fine mode particles in summer and autumn are similar to the increased concentration and oxidation process of DMS observed in previous studies. Therefore, we believe that the particles observed at Zhongshan Station may be related to DMS. In the discussion section, some air masses on low AOD days originated from the Antarctic plateau and may be associated with katabatic winds. Given the small particle sizes observed and referencing other literature on aerosol components from the Antarctic plateau, we infer that these particles may be non-sea-salt sulfates.

*"AOD675 nm is associated with the declining $\eta$"..... introduce $\eta$ before using it*

Respond:

Thank you for your careful review of our manuscript, which is helpful to improve the rigor of this manuscript! The following is our change in Section 3.2:

The solid black line represents the size of fine mode particles ($R_{eff}$), and the dashed blue line represents the proportion of the contribution of fine mode particles to AOD ($\eta$).

Finally, thank you again for your review of our manuscript and your valuable comments, which are of great help to our manuscript.

---

## Author Comment (AC2)

**Anonymous Referee #1:**

This study is too simple and may be well-suited as a measurement report. Undoubtedly, the aerosols in that location are very important to study, but the paper lacks enough depth to it.

However, I have some minor comments

Respond:

Thanks for your comments, we have changed the type of this manuscript to "measurement report". This manuscript does not analyze the AOD in detail, part of the reason is lacking of comprehensive observation. However, the CE318-T can provide unique information in the coastal Antarctic.

1. "The increase in AOD during spring and winter correlates with a reduction in the fine mode fraction, whereas the increase observed in summer and autumn may be attributed to the growth and aging of fine particles." How can both increase? Please correct

Respond:

What we meant is, "In winter and spring, high AOD values are related to the increase of coarse mode particles, while in summer and autumn, high AOD values may be related to the growth of fine mode particles." Detailed speaking, in Section 3.2, we use the aerosol classification method proposed by Gobbi to obtain the contribution of fine mode particles to AOD and the size of fine mode particles, and can also separate whether the increase of AOD is caused by the hygroscopicity growth of fine mode particles or by the increase of coarse mode particles. In Fig. 4, we discussed the different seasons. For example, in spring (Fig.4a), when AOD<0.03, the aerosol is mainly composed of fine mode particles ($\alpha_{440-870\,nm} > 1.0$), the contribution of fine mode particles to AOD is less than 70% ($\eta < 70\%$); when AOD>0.03, the aerosol is dominated by coarse mode particles ($\alpha_{440-870\,nm} < 1.0$), and the contribution of fine mode particles to AOD is less than 30% ($\eta < 30\%$). Moreover, since the particles during spring are mainly concentrated in the $\delta\alpha > 0$, this indicates that the increase of coarse mode particles is the main reason for the

increase of AOD in spring. In summer (Fig.4b), when AOD>0.03, aerosols mainly concentrated in the region of fine mode particle growth ($\delta\alpha < 0$), and the contribution of fine mode particles to AOD is 50% to 99% ($50\% < \eta < 99\%$), indicating a significant influence of fine mode particle growth on AOD increase. Therefore, we believe that the main reason for the higher AOD values is different during different seasons.

The former statement was unclear, which may lead to confusion for readers.

2. "Increases in AOD during spring and winter correlated with decreases in fine mode fraction, while increases during summer and winter related to fine mode particle growth and aging." This line is very confusing with the usage of 'increase''

Respond:

It has been modified into:

 "The high AOD values during winter and spring were associated with increased contribution of coarse mode particles, while high AOD values during summer and autumn are associated with the growth of fine mode particles."

3. The last line in abstract only talks about the origins of particles in the summer. Why specifically summer? Why not other seasons?

Respond:

Agree with you. In the discussion part, we used the air mass backward trajectory to analyze the sources of aerosol on high AOD days and low AOD days respectively. Although most of the high AOD days occur in summer, it is inaccurate to mention only summer in the abstract. Thus, it has been modified into:

"Backward trajectory analysis revealed that coarse particles from the ocean predominantly contributed to high AOD daily mean values, while fine particles on low AOD days originated mainly from the air mass over the Antarctic Plateau."

4. The abstract is incomplete. I suggest you to add a conclusion line to your abstract as to why this study is important or how it can help others?

Respond:

Agree. The abstract has been extended, such as the following is what we added in abstract:

This study enhances the understanding of the optical properties and seasonal behaviors of aerosols in the coastal Antarctic. Specifically, AOD measurements during the polar night address the lack of validation data for winter AOD simulations. Additionally, we revealed that lower wind speeds, higher temperatures, and lower relative humidity contribute to increased AOD at Zhongshan Station, and air masses from the ocean significantly impact local AOD levels. These findings help us infer AOD variation patterns in the coastal Antarctic based on meteorological changes, providing valuable insights for climate modeling in the context of global climate change.

5. How is DMS found in the plateau? Does it come from transportation from ocean? But you have mentioned about katabatic winds that drive from interior to coastal

Respond:

Thank you for your comments. In fact, we did not observe DMS particles in the Antarctic plateau. In Section 3.2, the seasonal variation in the proportion of fine particles aligns with the seasonal variation in DMS concentrations in previous studies, and the growth in fine mode particles observed during summer and autumn corresponds with the typical coastal process of DMS oxidation to MSA, nucleation, and growth. Therefore, we believe that the particles observed at Zhongshan Station may be related to DMS. In the discussion section, some air masses on low AOD days originated from the Antarctic plateau and may be associated with katabatic winds. Given the small particle sizes observed and referencing other literature on aerosol components from the Antarctic plateau, we infer that these particles may be non-sea-salt sulfates, primarily originating from DMS oxidation. Additionally, during summer, the enhanced efficiency of meridional long-range transport and the weakened inversion layer on the Antarctic Plateau likely facilitate the transport of these particles from the ocean to the Antarctic Plateau.

6. "AOD675 nm is associated with the declining $\eta$"….. introduce $\eta$ before using it

Respond:

The explanation of $\eta$ has been added in line 197-198:

"The solid black line represents the size of fine mode particles ($R_{eff}$), and the dashed blue line represents the proportion of the contribution of fine mode particles to AOD ($\eta$)."

---

## Author Comment (AC3)

The study offers a comprehensive analysis of AOD and AE variations at Zhongshan Station in Antarctica (where is unnoticed area in the community), providing valuable insights into seasonal and diurnal trends of these aerosol parameters. The authors have effectively utilized multiple data sources and analysis techniques to draw conclusions about the influence of meteorological factors, aerosol sources, and particle size dynamics on AOD and AE. Although the analysis looks simple, this research provide new in situ data and an important advancement on aerosol behavior in the Antarctic region, where the observation is very sparse and very hard to carry out. Thus I suggest acceptance after minor revision.

Respond:

Thank you very much for your positive evaluation and comments on our research. We greatly appreciate your valuable suggestions and will carefully consider them during the revision process. Your inputs will contribute to enhancing the quality and depth of our study, particularly in the challenging context of sparse observations in the Antarctic region. Our research provides new in situ data and represents an important advancement in understanding aerosol behavior. We will make revisions based on your suggestions.

**Major:**

1. The authors provide valuable insights into the relationship between particle size and AOD across different seasons. To strengthen the paper's cohesiveness, it would be beneficial to elaborate on the apparent seasonal differences in AOD variations. Specifically, clarifying how fine mode particles contribute to low AOD ranges while coarse mode particles are associated with high AOD values, and how this relationship evolves across seasons, would provide a more comprehensive understanding of the complex aerosol dynamics in the Antarctic environment.

Respond:

We agree that it is important to provide a more detailed explanation of the seasonal variations in AOD and clarify how fine mode particles contribute to low AOD ranges while coarse mode particles are associated with high AOD values.

The following is the modification we made in Section 3.1:

Although fine mode particles have a longer suspension time in the atmosphere and can efficiently scatter and absorb sunlight, leading to lower AOD ranges, it is worth mentioning that in the coastal regions of Antarctica, the dominant role in AOD is sometimes played by coarse mode particles. These particles, with larger radii and higher volume concentrations, originate mainly from abundant sea salt sources. Their presence results in increased scattering and absorption of sunlight, emphasizing the significance of coarse mode particles in determining AOD levels in the Antarctic coastal areas (Su et al., 2022).

The reference is:

Su, Y., Han, Y., Luo, H., Zhang, Y., Shao, S., and Xie, X.: Physical-Optical Properties of Marine Aerosols over the South China Sea: Shipboard Measurements and MERRA-2 Reanalysis, Remote Sensing, 14, 2453, https://doi.org/10.3390/rs14102453, 2022.

2. The study offers an intriguing analysis of the influence of wind speed on aerosol dispersal, noting that higher wind speeds generally lead to lower AOD values. However, the authors also highlight the role of blowing snow over sea ice in

generating sea salt aerosols, contributing to winter peaks in sea salt aerosols. To further enrich the discussion, it would be valuable to expand on the interplay between wind speed and AOD during winter, considering both the dispersal and production of aerosols. This additional context would provide a more nuanced understanding of the complex relationships at play in this unique environment.

Respond:

According to your comments, we investigated the relationship between wind direction and wind speed with AOD during the winter season at Zhongshan Station (Figure 1). The results indicate that higher AOD values are primarily associated with northeast and southeast winds, while lower AOD values are correlated with southwest winds, suggesting significant contributions from the ocean and marginal ice areas. However, the correlation coefficient between AOD and wind speed is relatively low (R = 0.078), with high AOD values observed across wind speeds ranging from 5 to 20 m/s. We acknowledge that the number of valid AOD observations during the winter season was limited, with only 98 data samples available for linear regression analysis with hourly wind direction and speed. This limitation may introduce considerable uncertainty, which is why we are not discussing winter now. If we want to obtain more reliable conclusions, we need to observe and accumulate enough valid data for a long time.

[Figure]

Figure 1. Relationship between AOD and wind direction and speed

3. Expanding the Discussion section to elaborate on how the observed AOD and AE variations relate to radiative forcing, cloud formation, and potential impacts on snow and ice melt in Antarctica. This would strengthen the study's significance by connecting the findings to broader climate-related processes.

Respond:

Agree. In the discussion section, we have added information about the interaction between Antarctic aerosol particles and clouds to highlight the importance of monitoring Antarctic aerosol optical properties.

4.2 Potential effects of aerosol particles on cloud and radiative forcing

The optical properties of aerosols play a crucial role in their impact on radiative forcing, cloud formation, and local climate. In our analysis of the variations in AOD and AE, we provided insights into the aerosol loading, particle sizes, and possible formation and growth mechanisms in the atmosphere over Zhongshan Station. During winter and spring, coarse mode particles are predominantly derived from sea salt. Studies have shown that aerosols larger than 0.13 $\mu m$ in the marine boundary layer contain sea salt, contributing to most of the aerosol scattering and inducing cooling effects (Murphy et al., 1998). Additionally, the size and inhomogeneity of sea salt particles are often associated with relative humidity. Compared to remote oceans, the low relative humidity in coastal Antarctica may introduce more inhomogeneous sea salt particles, resulting in up to a 12% change in direct radiative forcing due to inhomogeneity (Wang et al., 2019).

However, we are particularly interested in the behavior of aerosol particles during summer since solar radiation is limited in winter. In summer and autumn, the increase in fine mode particles in closely related to the release of biogenic aerosols, such as DMS, emitted by phytoplankton in the marginal ice zone. When particles grow to a size suitable for cloud condensation nuclei or ice nucleating particles, they can affect the formation of low-level mixed-phase clouds in coastal areas, contributing to the formation of low-level ice clouds. At the same time, the increased number density of cloud droplets enhances cloud reflectivity, resulting in negative radiative forcing (Satheesh and Krishna Moorthy, 2005). A recent study revealed that in the shallow

mixed-phase clouds over Antarctica, the concentrations of cloud-relevant aerosol particles match the concentrations of ice crystals and cloud droplets (Radenz et al., 2024). the number of particles plays a crucial role in cloud growth. Increasing particle concentration results in a higher abundance of liquid droplets and ice crystals within clouds, which can impact cloud lifespan and potentially influence local weather and climate. Therefore, continuous monitoring of aerosol optical properties in coastal Antarctica is vital to improve our comprehension of aerosol radiative forcing variations caused by changes in aerosol loading and particle size.

The newly added references are:

Murphy, D. M., Anderson, J. R., Quinn, P. K., McInnes, L. M., Brechtel, F. J., Kreidenweis, S. M., Middlebrook, A. M., Pósfai, M., Thomson, D. S., and Buseck, P. R.: Influence of sea-salt on aerosol radiative properties in the Southern Ocean marine boundary layer, Nature, 392, 62–65, https://doi.org/10.1038/32138, 1998.

Radenz, M., Engelmann, R., Henning, S., Schmithüsen, H., Baars, H., Frey, M. M., Weller, R., Bühl, J., Jimenez, C., Roschke, J., Muser, L. O., Wullenweber, N., Zeppenfeld, S., Griesche, H., Wandinger, U., and Seifert, P.: Ground-based Remote Sensing of Aerosol, Clouds, Dynamics, and Precipitation in Antarctica —First results from the one-year COALA campaign at Neumayer Station III in 2023, https://doi.org/10.1175/BAMS-D-22-0285.1, 2024.

Satheesh, S. K. and Krishna Moorthy, K.: Radiative effects of natural aerosols: A review, Atmospheric Environment, 39, 2089–2110, https://doi.org/10.1016/j.atmosenv.2004.12.029, 2005.

Wang, Z., Bi, L., Yi, B., and Zhang, X.: How the Inhomogeneity of Wet Sea Salt Aerosols Affects Direct Radiative Forcing, Geophysical Research Letters, 46, 1805–1813, https://doi.org/10.1029/2018GL081193, 2019.

4. Addressing the potential impact of missing measurements and instrument downtime on the correlation analysis of AOD and AE with meteorological variables. Quantifying measurement uncertainty and discussing its implications for the interpretation of correlation coefficients would further enhance the study's rigor.

Respond:

In the data preprocessing stage, we systematically removed invalid data caused by instrument downtime, daily observations fewer than 20, and low solar elevation angles. However, according to the estimation of AOD uncertainty for CE318-T by Barreto et al. (2016), during daytime, the uncertainty primarily stems from the calibration term, with the field instrument AOD standard uncertainty ranging from ~0.015. For nighttime measurements, the AOD uncertainty depends on the calibration technique used. Specifically, when calibrated using the Moon Ratio technique, the AOD uncertainty ranges from 0.011 to 0.019. However, if the new Sun Ratio technique is applied, higher uncertainties are expected, specifically 0.012 to 0.015 (0.017) for the visible (440 nm) channels and 0.015 to 0.021 for longer wavelengths. Additionally, for instruments calibrated using the new Sun-Moon gain factor technique and using a Langley-calibrated instrument for G calculation, the uncertainties range from 0.016 to 0.019. We will include an explanation of the on-site CE318-T AOD uncertainty in Section 2.2.

The following is the modification we made in Section 2.2:

It should be noted that there are uncertainties in the AOD measurements of CE318-T during field observations due to atmospheric conditions, instrument noise, and calibration. It is estimated that during daytime measurements, the AOD uncertainty ranges from 0.010 to 0.021. For night-time measurements, the AOD uncertainty depends on the calibration technique used. Specifically, when calibrated using the Moon Ratio technique, the uncertainty ranges from 0.011 to 0.019. With the application of the new Sun Ratio technique, the uncertainty for the 440 nm channel is between 0.012 and 0.015 (0.017), while for longer wavelengths, it ranges from 0.015 to 0.021. By employing the new Sun-Moon gain factor technique and using the Langley-calibrated instrument for calculation of the amplification between daytime and night-time measurements, the uncertainty range is from 0.016 to 0.019(Barreto et al., 2016).

The reference is:

Barreto, Á., Cuevas, E., Granados-Muñoz, M.-J., Alados-Arboledas, L., Romero, P. M., Gröbner, J., Kouremeti, N., Almansa, A. F., Stone, T., Toledano, C., Román, R., Sorokin, M., Holben, B., Canini, M., and Yela, M.: The new sun-sky-lunar Cimel CE318-T

multiband photometer - a comprehensive performance evaluation, Atmospheric Measurement Techniques, 9, 631–654, https://doi.org/10.5194/amt-9-631-2016, 2016.

**Minor:**

1. Line 54-55, line 122, and line 314: when describing the values of AOD and AE, it is suggested to unify the number of decimal places to improve accuracy.

Respond:

Thank you for your detailed review. We will follow your suggestion to unify the number of decimal places for the values of AOD and AE mentioned in lines 54-55, 122, and 314 to improve accuracy. The following is our changes in manuscript:

Line 54-55: AOD observation records from Antarctica sites indicate that the values range from 0.006 to 0.220 in coastal regions and from 0.007 to 0.034 in inland regions.

Line 122: The monthly mean AOD values at 500 nm ($AOD_{500\,nm}$) generally remained below 0.10, consistent with findings by Gadhavi and Achuthan at the Maitri Station, where AOD variation fell within the range of 0.01 to 0.10 (Gadhavi and Achuthan, 2004).

Line 314: A weak positive correlation was noted between temperature and AOD (R = 0.22, p = 0.40), and a negative correlation between relative humidity and AOD (R = -0.59, p = 0.02).

2. It is suggested to retain three decimal places for the values in Table 1 to ensure consistency with the number of decimal places used in the manuscript.

Respond:

Thank you for your detailed review and valuable feedback on our manuscript. We will follow your suggestion to retain three decimal places for the values in Table 1 to ensure consistency with the number of decimal places used in the manuscript. The following is our change:

|  | 2020 | 2021 | 2022 | 2023 |
|---|---|---|---|---|
| $AOD_{1640\,nm}$ | 0.028±0.102 | 0.026±0.079 | 0.050±0.141 | 0.016±0.036 |

| | | | | |
|---|---|---|---|---|
| AOD$_{1020 nm}$ | 0.049±0.095 | 0.045±0.073 | 0.067±0.131 | 0.040±0.034 |
| AOD$_{870 nm}$ | 0.047±0.093 | 0.039±0.070 | 0.060±0.126 | 0.037±0.032 |
| AOD$_{675 nm}$ | 0.059±0.091 | 0.042±0.068 | 0.063±0.122 | 0.044±0.031 |
| AOD$_{500 nm}$ | 0.074±0.090 | 0.051±0.066 | 0.071±0.117 | 0.053±0.031 |
| AOD$_{440 nm}$ | 0.081±0.089 | 0.057±0.065 | 0.077±0.116 | 0.057±0.031 |
| AOD$_{380 nm}$ | 0.089±0.091 | 0.063±0.065 | 0.077±0.117 | 0.061±0.032 |
| AOD$_{340 nm}$ | 0.088±0.095 | 0.059±0.064 | 0.073±0.118 | 0.058±0.032 |
| AE$_{440-870 nm}$ | 1.134±0.411 | 0.953±0.338 | 0.883±0.374 | 0.753±0.206 |

3. 3. There is an error in the reference of Figure 5 in line 224, and it is suggested to be corrected as (Fig.5).

Respond:

Thank you for your detailed review, we have corrected it. The following is our change:

Line 225: …with each hourly mean containing at least one thousand individual observations (Fig. 5).

4. Please correct the notation of Celsius (℃) in line 84 and Figure 7.

Respond:

Thank you for your detailed review, we have corrected it. The following is our change:

Line 84: The average annual air temperature is -10 ℃, with a relative humidity of 58% and prevailing wind speeds of 6.9 m s$^{-1}$, primarily from the east or east-southeast direction (Ding et al., 2022).

Figure 7:

[Figure]

5. In Section 3.2, the authors can make a more natural transition from a discussion of high concentrations of sea salt aerosols in winter as the cause of high AOD to the discussion of DMS and MSA in summer. It is suggested to make some adjustments to the statement.

Respond:

Thank you for your comment. To ensure a more natural transition, we have modified it to:

"In summer, lower sea salt concentrations lead to lower background levels of AOD, but the effect of enhanced marine biogenic emissions on AOD may increase."

6. When describing Figure 4, the author should explain what the parameters in the figure represent, such as and, it helps the readers understand the following analysis.

Respond:

The explanation of the parameters $R_{eff}$ and $\eta$ has been added in line 197-198:

"The solid black line represents the size of fine mode particles ($R_{eff}$), and the dashed blue line represents the proportion of the contribution of fine mode particles to AOD ($\eta$)."

7. In Section 3.3, the authors discussed the relationship between temperature and relative humidity and the diurnal variation of AOD. Is there a physical mechanism to explain the positive or negative impact of temperature and relative humidity changes on aerosol load at Zhongshan Station?

Respond:

At present, there is limited research on the mechanism of local temperature affecting aerosol load in Antarctica. However, by referring to relevant literature exploring the impact of meteorological factors on AOD in the mid-latitudes, we can find a general positive correlation between temperature and AOD. This correlation is attributed to the fact that higher temperatures are conducive to the generation of particulate matter. While our manuscript discusses the potential impacts of temperature and relative humidity on AOD at Zhongshan Station, further research is needed to thoroughly understand the detailed physical mechanisms involved.

8. There are still some grammatical errors in the manuscript. Please revise carefully. For example:

Line 99-100: change "eliminate" to "eliminated" for correct tense. Change "exceeding" to "exceedingly" for correct adverb form.

Respond:

It has been modified into:

"Consequently, we categorize daily observations with less than 20 measurements and the coefficient of dispersion (CV) exceeding 1 as invalid data, which are systematically eliminated from our analysis. Typically, these invalid data manifest with exceedingly high AOD values, often attributed to instrument downtime caused by factors such as precipitation or cloudy weather."

*Line 225-226: "from 5:00 to 12:00 to the lowest value" can be changed to "from 5:00 to 12:00, reaching the lowest value."*

Respond:

It has been modified into:

"The mean $AE_{440-870\,nm}$ decreased from 5:00 to 12:00, to reaching the lowest value (0.85 ±0.25), and then increased."

*Line 240: "average speeds range from 2 to 9 m s$^{-1}$" should be changed to "average speeds ranging from 2 to 9 m s$^{-1}$".*

Respond:

It has been modified into:

"Moreover, the diurnal variation of the 2-minute wind at Zhongshan Station reveals prevailing southeast direction, with average speeds ranging from 2 to 9 m s$^{-1}$."

*Line 252: "by influencing the air convection and influences the formation" should be "by influencing the air convection and influencing the formation".*

Respond:

It has been modified into:

"Temperature affects aerosol particle concentration by influencing the air convection and influencing the formation and optical properties of secondary by controlling chemical transformation (Li et al., 2020; Han et al., 2007)."

9.  Some expressions in the manuscript could be further streamlined to enhance the quality of the article. For example:

Line 126-129: The statements "The annual mean ± SD values of the AOD500 nm were 0.074±0.090, 0.051±0.066, 0.071±0.117, and 0.053±0.031 in 2020, 2021, 2022, and 2023, respectively (Table 1)" and "The annual mean ± SD values of the AE440-870 nm

were 1.134±0.411, 0.953±0.338, 0.883±0.374, 0.753±0.206 in 2020, 2021, 2022, and

2023" can be combined into one sentence to reduce redundancy.

Respond:

It has been modified into:

"The annual mean ± SD (standard deviation) values of the $AOD_{500\ nm}$ were 0.074±

0.090, 0.051±0.066, 0.071±0.117, and 0.053±0.031 in 2020, 2021, 2022, and 2023,

respectively (Table 1). Similarly, the annual mean ± SD values of the $AE_{440-870\ nm}$ were

1.134±0.411, 0.953±0.338, 0.883±0.374, and 0.753±0.206 for the same years,

respectively, suggesting that the aerosols over Zhongshan Station were mainly

dominated by fine mode particles in 2020 and by coarse mode particles in 2021, 2022,

and 2023."

Line 270-280: "In addition to meteorological conditions that can affect the diurnal

variation characteristics of AOD, we believe that aerosol sources may be another

influencing factor" can be simplified to "Besides meteorological conditions, aerosol

sources may also influence the diurnal variation characteristics of AOD".

Respond:

It has been modified into:

"Besides meteorological conditions, aerosol sources may also influence the diurnal

variation characteristics of AOD."

10. The summary section should not rehash the detailed reasons for seasonal variations
    in AOD and AE, as these have been discussed in the results section. Simplification
    is recommended.

Respond:

The summary section has been simplified to:

At Zhongshan Station, AOD varied from 0.00 to 0.20. Fine mode particles were

predominantly found in the lower AOD range, while higher AOD values were mainly

attributed to coarse mode particles. Seasonally, AOD exhibited a pattern of lower values in summer and higher values in winter, and the AE displayed an opposite trend. The increases in AOD during summer and autumn may be linked to particle growth, whereas the increases during spring and winter are associated with a decline in the fraction of fine mode particles.

---

## Author Response (AR2)

**Report #1:**

1. What is 2-minute wind?

Respond:

The 2-minute wind we referred to in the manuscript actually signifies the average wind speed and direction measured over a two-minute period. To ensure precision in terminology, we will modify it to "2-minute average wind speed" and "2-minute average wind direction". We appreciate your careful review, which will help improve the accuracy and clarity of the manuscript.

The following is the modification we made in Section 3.3:

"Moreover, the 2-minute average wind direction at Zhongshan Station mainly comes from the southeast, with the diurnal variation of the 2-minute average wind speeds ranging from 2 to 9 $m\ s^{-1}$."

"Figure 7 Diurnal variations of 2-minute average wind direction, 2-minute average speed, and $AOD_{500\ nm}$ in summer at Zhongshan Station."

2. Are there any studies in the Arctic or other pristine sites with similar absorption and concentration of aerosols where they didn't find aerosol boundary layer positive feedback?

Respond:

We acknowledge that there is currently a relative lack of studies on the positive feedback between low-concentration aerosols and the boundary layer in the Arctic or other pristine sites. This is because significant aerosol-boundary layer feedback typically occurs in heavily polluted urban areas where the concentration of absorbing aerosols is high. Although some studies have observed higher concentrations of absorbing aerosols in the upper troposphere during the Arctic summer, there has been insufficient discussion regarding their feedback with the boundary layer (Igel et al., 2017; Ansmann et al., 2023). Additionally, due to the limited sources of absorbing aerosols and the lower aerosol concentrations in the Antarctic region, there may be a lack of studies that can serve as references.

The references are:

Ansmann, A., Ohneiser, K., Engelmann, R., Radenz, M., Griesche, H., Hofer, J., Althausen, D., Creamean, J. M., Boyer, M. C., Knopf, D. A., Dahlke, S., Maturilli, M., Gebauer, H., Bühl, J.,

Jimenez, C., Seifert, P., and Wandinger, U.: Annual cycle of aerosol properties over the central Arctic during MOSAiC 2019–2020 – light-extinction, CCN, and INP levels from the boundary layer to the tropopause, Atmospheric Chemistry and Physics, 23, 12821–12849, https://doi.org/10.5194/acp-23-12821-2023, 2023.

Igel, A. L., Ekman, A. M. L., Leck, C., Tjernström, M., Savre, J., and Sedlar, J.: The free troposphere as a potential source of arctic boundary layer aerosol particles, Geophysical Research Letters, 44, 7053–7060, https://doi.org/10.1002/2017GL073808, 2017.

3. Line 266: $AOD_{500\ nm}$ exhibits a decline (increase) concurrent with decreasing (increasing) wind speeds. Shouldn't AOD decrease with increasing wind speed?

Respond:

Thank you for pointing out the error in our manuscript. We sincerely apologize for this mistake and have corrected it to: "$AOD_{500\ nm}$ exhibits a decline (increase) concurrent with increasing (decreasing) wind speeds."

4. How does the boundary layer vary with temperature? Doesn't convection have any impact on boundary layer in Antarctica?

Respond:

From Figure 1, it is evident that the diurnal variation characteristics of boundary layer height and temperature at Zhongshan Station are similar. Generally, the boundary layer height is primarily influenced by the diurnal cycle of energy budgets for both the Earth's surface and atmosphere, which typically leads to peak boundary layer heights occurring in the afternoon. However, we observe a slight decrease in boundary layer height during the period from 13:00 to 14:00. This suggests that there may be other factors affecting boundary layer height beyond solar radiation. We guess this phenomenon may be related to enhanced convective activity and lower wind speeds around noon, which facilitate the generation and accumulation of surface particles, thus resulting in a decrease in boundary layer height.

[Figure]

**Figure 1 the relationship between boundary layer height and temperature at Zhongshan Station.**

5. Can you also check precipitation along with high rh? This might explain scavenging effect which reduces AOD in summer?

Respond:

Thank you very much for your insightful suggestion. We agree that precipitation along with high relative humidity could be a significant process for aerosol scavenging during summer. However, since the CE318-T ceases observations during overcast or precipitation events, it is challenging for us to accurately assess the changes in AOD during these periods. Perhaps utilizing satellite AOD products could provide better insights into the impact of precipitation on aerosol scavenging. We will consider this approach for further investigation.

6. Can you include a figure of the backward trajectories season wise? It would be interesting to learn if you are considering only natural/background aerosols or long-range transport aerosols too in the AOD study. It would be nice if you divided the data into local vs long range transport and see how AOD behaves.

Respond:

Thank you for your valuable suggestions. we calculated the 168h backward trajectory once every 24h from January 2020 to December 2022, with the starting height of 500m and the starting time of 18:00 (13:00 local time at Zhongshan), and clustered by season (Figure 2). As you pointed out, the clustering results clearly indicate that the red and blue clusters are likely associated with local aerosol sources, while the yellow and green clusters are related to long-range aerosol sources.

We found that the contribution of local aerosols dominates throughout the year, while the influence of long-range transport aerosols is more pronounced in the autumn compared to other seasons. This suggests that the seasonal differences in AOD and AE at Zhongshan Station are largely driven by variations in the types and concentrations of local aerosols.

[Figure]

**Figure 2 Clusters of air mass backward trajectories in (a) spring, (b) summer, (c) autumn and (d) winter at Zhongshan Station from 2020 to2022.**

The following is the modification we made in Section 3.2:

To investigate the seasonal differences in AOD and AE, it is essential to understand the sources of air masses influencing aerosols at Zhongshan Station. Therefore, we calculate the 168h backward trajectory once every 24h from January 2020 to December 2022, with the starting height of 500 m and the starting time of 18:00 (13:00 local time at Zhongshan Station), and clustered by season (Fig. 3). We observed that the proportion of air masses originating from the surrounding waters (red clusters) and the ice edge margin of ice sheet (blue clusters) ranges from 50% to 80%, dominating throughout the year. These air masses are likely the primary sources of local or natural aerosols in Antarctica. In contrast, the proportion of distant origin (yellow and green clusters) is approximately 20%, but it significantly increases in autumn, reaching around 45%. These air masses are associated with long-range transported aerosols. Therefore, the seasonal differences in AOD and AE at Zhongshan Station are largely attributed to variations in the types and concentrations of local aerosols.

7.  Can you extend the correlation with meteorological factors to other seasons? why particularly summer?

Respond:

Thank you for your comments. The main reasons we focus on summer are the following:

1) Data quantity consideration: During the summer, we obtained more AOD observations data (with over 1000 measurements per hour from 5:00 to 20:00), which enhances the reliability of the data. However, during other seasons, particularly winter, the amount of effective data after quality control is relatively low, we consider that these data may lack seasonal representativeness, which would introduce considerable uncertainty in the subsequent analysis of the correlation between AOD and meteorological factors.

2) Time range of the observation: The lower solar zenith angle in summer allows for a wider time range for studying diurnal variation of AOD.

3) Further study: We believe that the ultimate goal of studying the optical properties of aerosols in Antarctica is to understand their radiative characteristics. Since summer offers enough solar radiation while winter lacks solar radiation, we opted to focus on the summer period for our research.